# New Hydrophilic Matrix Tablets for the Controlled Released of Chlorzoxazone

**DOI:** 10.3390/ijms25105137

**Published:** 2024-05-09

**Authors:** Andreea Creteanu, Gabriela Lisa, Cornelia Vasile, Maria-Cristina Popescu, Daniela Pamfil, Claudiu N. Lungu, Alina Diana Panainte, Gladiola Tantaru

**Affiliations:** 1Department of Pharmaceutical Technology, Faculty of Pharmacy, “Grigore T. Popa” University of Medicine and Pharmacy, 16 Universitatii Street, 700115 Iași, Romania; acreteanu@gmail.co; 2Department of Chemical Engineering, Faculty of Chemical Engineering and Environmental Protection, “Gheorghe Asachi” Technical University, 73 Dimitrie Mangeron Prof., Str., 700050 Iași, Romania; gapreot@yahoo.com; 3Physical Chemistry of Polymers Department, Petru Poni Institute of Macromolecular Chemistry, 41A Gr. Ghica Voda Alley, 700487 Iași, Romania; cvasile@icmpp.ro (C.V.); cpopescu@icmpp.ro (M.-C.P.); pamfil.daniela@icmpp.ro (D.P.); 4Departament of Functional and Morphological Science, Faculty of Medicine and Pharamacy, Dunarea de Jos University, 800008 Galati, Romania; 5Department of Analytical Chemistry, Faculty of Pharmacy, “Grigore T. Popa” University of Medicine and Pharmacy, 16 Universitatii Street, 700115 Iași, Romania; gtantaru2@yahoo.com

**Keywords:** chlorzoxazone, kollidon, chitosan, matrix tablets, controlled release

## Abstract

The modified release of active substances such as chlorzoxazone from matrix tablets, based on Kollidon^®^SR and chitosan, depends both on the drug solubility in the dissolution medium and on the matrix composition. The aim of this study is to obtain some new oral matrix tablet formulations, based on Kollidon^®^SR and chitosan, in order to optimize the low-dose oral bioavailability of chlorzoxazone, a non-steroidal anti-inflammatory drug of class II Biopharmaceutical Classification System. Nine types of chlorzoxazone matrix tablets were obtained using the direct compression method by varying the components ratio as 1:1, 1:2, and 1:3 chlorzoxazone/excipients, 20–40 w/w % Kollidon^®^SR, 3–7 w/w % chitosan while the auxiliary substances: Aerosil^®^ 1 w/w %, magnesium stearate 0.5 w/w % and Avicel^®^ up to 100 w/w % were kept in constant concentrations. Pharmaco-technical characterization of the tablets included the analysis of flowability and compressibility properties (flow time, friction coefficient, angle of repose, Hausner ratio, and Carr index), and pharmaco-chemical characteristics (such as mass and dose uniformity, thickness, diameter, mechanical strength, friability, softening degree, and in vitro release profiles). Based on the obtained results, only three matrix tablet formulations (F1b, F2b, and F3b, containing 30 w/w % KOL and 5 w/w % CHT, were selected and further tested. These formulations were studied in detail by Fourier-transform infrared spectrometry, X-ray diffraction, thermogravimetry, and differential scanning calorimetry. The three formulations were comparatively studied regarding the release kinetics of active substances using in vitro release testing. The results were analyzed by fitting into four representative mathematical models for the modified-release oral formulations. In vitro kinetic study revealed a complex mechanism of release occurring in two steps of drug release, the first step (0–2 h) and the second (2–36 h). Two factors were calculated to assess the release profile of chlorzoxazone: f1—the similarity factor, and f2—the factor difference. The results have shown that both Kollidon^®^SR and chitosan may be used as matrix-forming agents when combined with chlorzoxazone. The three formulations showed optima pharmaco-technical properties and in vitro kinetic behavior; therefore, they have tremendous potential to be used in oral pharmaceutical products for the controlled delivery of chlorzoxazone. In vitro dissolution tests revealed a faster drug release for the F2b sample.

## 1. Introduction

Sustained drug delivery systems try to regulate the drug release rate to maintain targeted blood medication levels that are therapeutically effective for a prolonged time. Lowering the overall dose of the medication given and the frequency of adverse side effects can, therefore, improve patient compliance. Drug loading into a matrix system is the most popular way to control its release [1,2]. A centrally acting muscle relaxant, chlorzoxazone (5-chloro-2,3-dihydro-1,3-benzoxazol-2-one) (CLZ) is considered a Class II drug (according to the Biopharmaceutical Classification System, BCS) because it has low solubility and high membrane permeability. CLZ has been licensed for treating musculoskeletal disorders [3] and is used to relieve muscle spasms and subsequent pain and discomfort [4]. By predominantly acting at the level of the spinal cord and subcortical regions of the brain, CLZ prevents muscle spasms. CLZ is a centrally-acting musculoskeletal relaxant with sedative properties. It constrains the multisynaptic reflex arcs produced from the spinal cord and subcortical regions of the brain, which prolongs the cause for maintaining the skeletal muscle spasm. CLZ is also used for acute pain relief and headache due to muscle contraction [5].

Inhibiting calcium and potassium influx results in muscular relaxation and neural inhibition. Following oral dosing, mice and rats tested on CLZ showed little to no muscle relaxant activity, and the drug was fully absorbed and quickly metabolized in the liver to 6-hydroxychlorzoxazone [4,6]. It acts an hour after an oral dose and lasts for three to four hours. For most tested participants, CZT may attain peak levels after 1 to 2 h following oral administration, with the highest CLZ blood level typically noticed in persons during the first 30 min. Therapeutically active plasma concentrations are maintained for 3–4 h. The mean plasma half-life is 1 h. CLZ is rapidly metabolized in the liver by the CYP2E isozyme of the citrochrome P450. CLZ is renally eliminated, mainly in conjugated form (as glucuronide) and <1% in an unchanged form [7].

The recommended starting oral dosage is 500 mg three or four times per day; however, this can frequently be lowered to 250 mg three or four times per day in the future. CLZ is typically administered alongside analgesics. After being quickly digested, CLZ is mainly eliminated as the conjugated form of glucuronide in the urine. Within 24 h, the urine excretes less than 1% of the administered dose of CLZ undisturbed. The three most typical side effects of CLZ are headache, lightheadedness, and drowsiness [8]. Rarely, there have been reports of severe—even fatal—hepatocellular toxicity with CLZ treatment. Although the process is unknown, it seems peculiar and erratic; it is unknown what the factors are that predispose persons to this rare condition. Early hepatotoxicity signs and/or symptoms, such as fever, rash, anorexia, nausea, vomiting, lethargy, right upper quadrant pain, dark urine, or jaundice, should be reported by patients [9]. CLZ has a Pka value of 3.3, a lower aqueous solubility of 0.2–0.3 mg/mL, and high membrane permeability; therefore, dissolution in gastrointestinal fluids is the limiting step for its oral bioavailability [10]. Chlorzoxazone is rapidly absorbed from the gastrointestinal tract. Therapeutically active plasma concentrations are maintained for 3–4 h.

The solubility of a drug influences the choice of formulation for oral administration, dissolution, and absorption in the digestive tube and whether it is a suitable candidate for formulation as a gastroretentive dose form. Longer stomach transit times contribute to increased solubility and, thus, to absorption. Water-insoluble medicines have limited therapeutic effects due to low solubility and dissolving rates [11]. The gastrointestinal absorption of any active substance is severely limited by its low water solubility. In earlier research, we looked at different methods to improve the active ingredient’s solubility and the impact of formulation parameters on CLZ’s stability [12,13,14]. The study is predicated on novel oral matrix tablets that were designed to maximize the low oral bioavailability using chitosan and Kollidon^®^SR. Made up of 80% polyvinyl acetate (Mr = 450,000 Daltons) and 20% polyvinylpyrrolidone (povidone) (Mr = 40,000 Daltons), Kollidon^®^SR (KOL) is a physical mixture of polymers [15,16]. KOL is one of the utilized hydrophilic excipients in the formulation and production of modified-release matrix tablets. KOL is presented as a white or yellowish-white powder with spherical particles of 80–100 μm (Figure 1), insoluble in water, and very stable (more than two years) at 20–25 °C. It has flow properties, an angle of repose less than 30°, an angle of repose of 21°, a flow time of 9.5 g/s, a compression force of 10 kN, and a hardness of 170 N. The literature contains a wealth of research showcasing that matrix-forming agents are effective and adaptable. The release of the active substance from the matrix modified-release tablets is dependent on its degree of solubility in the dissolution medium as well as on the composition of the matrix forming. Analogous research was published in the literature [17] (Figure 2).

Chitosan (CHT) has also been used in studied KOL-based formulations. The chemical structure of CHT (C_6_H_11_O_4_N) includes two co-monomers, glucosamine and N-acetyl glucosamine, being considered as a chitin/chitosan copolymer with formulae given in Figure 3. It has an average high molecular weight of about Mw = 10,000–1,000,000 Daltons, with a degree of deacetylation between 40–98%. CHT is a biodegradable and biocompatible polymer that acts as an absorption promoter for hydrophobic active substances with high molecular weight in the gastrointestinal tract [18,19].

In this study, CHT with a degree of deacetylation from 51% to 65% was used because only such kind of polymer increases the absorption of active hydrophobic substances with high molecular weight of 10.000–1.000.000 Daltons [20].

Other materials used were: microcrystalline cellulose as Avicel^®^ PH (AV): Aerosil^®^ (A) as hydrophilic fumed silica with a specific surface area of 200 m² g^−1^ and magnesium stearate (ST). All used compounds accomplish the quality requirements according to the laws of force. These excipients facilitate the application of the direct compression method to obtain optimal dispersibility of the hydrophobic substances (CLZ) in the powdered mixtures and the association of hydrophilic (CHT).

The objective of the study was to develop and pharmacologically characterize the matrix tablets based on Kollidon^®^SR and chitosan, formulated to optimize the oral availability of CLZ, taking into account the pharmacodynamic and biopharmaceutical properties of this muscle relaxant drug. It is known that it is necessary to develop and implement new delivery systems that combine safety and efficacy to improve the solubility of CLZ, either by increasing its oral bioavailability, reducing its adverse effects, or reducing the number of doses to improve patient compliance. The new oral matrix tablets with CLZ based on Kollidon^®^SR and chitosan were developed as an alternative to oral solid dosage forms for pediatric and geriatric patients who have difficulties swallowing solid dosage forms. The present matrix tablets for pediatric and geriatric practice versus traditional tablets aim to improve patient compliance, reduce the number of administered doses, reduce adverse effect, and increase solubility by preparing sustained-release tablets.

## 2. Results

The compositions of the studied formulations are given in Table 1.

Determination of the flow and compressibility parameters of the powder mixtures was undertaken on mixtures of powders for the nine proposed formulations, using CLZ 50, 33, and 25 w/w %, KOL 30–50 w/w %, CHT 3–7 w/w %, ST, AE as a constant value, and the AV completes up to 100 w/w %. The influence of KOL and CHT on the quality of the matrix and controlled drug release were followed.

The results obtained by the determination of the flow and compressibility parameters of the studied formulations are presented in Table 2. The influence of KOL and CHT on the quality of the matrix was followed [15,21]. All the flow parameters determined showed a positive influence of CHT on the flowability and compressibility properties. A good flow of formulations (F1a, F1b, F2a, F2b, F3a, and F3b) with 20–30 KOL w/w % concentration was evidenced. In contrast, the increase in KOL concentration above 40 w/w % negatively influenced the flow properties; thus, the F1c, F2c, and F3c formulations as powders showed a deficient flow. The results obtained for the Hausner ratio and Carr index set the F1a, F1b, F2a, F2b, F3a, and F3b formulations into the group of powders with sound and excellent flow.

It was also found that increasing the CHT concentration in the formulation resulted in a decrease in the rate of CLZ release from the matrix tablets (see below).

The results obtained for the pharmaco-technical parameters of matrix tablet formulations are shown in Table 3.

The analysis of the obtained values proved that the F1c, F2c, and F3c formulations containing the highest percentage of KOL exhibited significant variations in mass uniformity, even beyond the 5% limit set by European Pharmacopoeia, 8th ed. [22]. It could also be correlated to the data recorded for the flow and compressibility parameters that indicated a poor flow for those formulations. Formulations F1a,b, F2a,b, and F3a,b showed variations in tablet mass within the limits set by European Pharmacopoeia, 8th ed. [22].

Mechanical strength varied between 82.45 N and 99.75 N, and it was also noticed that the values decreased directly proportionally to the increase in the concentration of KOL. Those results are not by some technical specifications for KOL, which stated an increase in the mechanical strength of the tablets directly proportional to the increase of the polymer concentration of the formulation as a result of its plastic behavior imprinted by the povidone-linked polyvinyl acetate [15]. That behavior with regard to compression has also been observed in other studies [18] when KOL was combined with other hydrophilic polymers as matrix components.

The mass uniformity shows a deviation of ±5% compared to the mass of standard tablets of ≥250 mg (500 ± 25 mg). The dose uniformity values correspond to each unit’s individual content of active substance, which should be 95% to 105% as the average content.

Data in Table 3 shows that the average drug content of three tablets from each formula was 249 mg for F1, 165.66 mg for F2, and 124.67 mg for F3. The standard deviation (SD) values indicate that the sample processing made no significant modifications. All obtained values of the pharmaco-technical characteristics are within the limits of the specifications reported in the literature [15].

CLZ was quantitatively determined using an HPLC method. It was detected by the UV spectrum at its characteristic wavelength of 280.4 nm [13] (Figure 4).

The HPLC chromatogram highlights the quantitative difference in the active substance from the three formulations. F1 has the highest peak, containing the highest amount of CLZ, followed by F2 with the smallest peak and F3 with the lowest peak, indicating the least CLZ titrates.

Friability, a pharmaco-technical parameter directly correlated with the mechanical strength of the tablets, showed increasing values directly proportional to the increase in the concentration of KOL in the formulation. As far as the friability was concerned, it was noticed that the decrease in mechanical strength had led to an increase in the friability of the tablets. The F1c, F2c, and F3c formulations showed a stripping tendency during compression and friability tests.

The hydration characteristics of the hydrophilic matrix tablets with CLZ are plotted in Figure 5.

The matrix-forming polymers directly influenced the hydration characteristics of hydrophobic matrix tablets with modified release. Thus, F1a, F1b, F1c, F2a, F2b, F2c, F3a, F3b, and F3c formulations with 20–40 w/w % KOL concentration exhibited absorbent properties in the first five hours of the test, after that, in the following 2–3 h there was a slight decrease in mass. From the 8th hour, the matrix tablets were almost unchanged in size and mass. It is worth mentioning that formulations F1b, F2b, and F3b with 30 w/w % KOL showed hydration properties superior to other formulations depending on the content of the KOL and CHT. Formulations F1c, F2c, and F3c with 40 w/w % KOL and 7 w/w % CHT were characterized by adsorbent properties inferior to the other formulations [21].

Only three formulations, F1b, F2b, and F3b, showed optimum pharmaco-technical properties. They were characterized in detail by Fourier-transform infrared spectrometry (FT-IR), X-ray diffraction, thermogravimetry (TG), differential scanning calorimetry (DSC), and in vitro dissolution tests.

### 2.1. Drug-Excipients Compatibility Study

#### 2.1.1. Fourier Transform Infrared Spectroscopy

FT-IR spectroscopy has been performed to assess the compatibility of the components of the formulations. Figure 6 presents the FT-IR spectra of all components used in the formulation preparation process.

As can be seen from Figure 6, the IR spectrum of CLZ [23] shows bands at 3468, 3204, and 3156 cm^−1^ assigned to stretching vibration of NH groups, at 3080 and 3055 cm^−1^ assigned to stretching vibration of Ph-H groups, at 2982 and 2827 cm^−1^ assigned to stretching vibration of CH groups, at 1773 cm^−1^ assigned to stretching vibration of C=O groups, at 1614 cm^−1^ assigned to C=C groups from aromatic rings, at 1471 cm^−1^ assigned to the stretching vibration of C-C and CN groups and deformation vibration of CCH groups, at 1359 and 1255 cm^−1^ assigned to stretching vibration of CC groups and deformation vibration of CCH and CNH groups, at 1100 cm^−1^ assigned to stretching vibration of CN and CO groups and deformation vibration of CCH and CCC groups, at 1061 cm^−1^ assigned to the stretching vibration of CC and CCl groups and deformation vibration of CCH groups, at 958 cm^−1^ assigned to the stretching vibration of CN groups and deformation vibration of CCC groups, at 591 cm^−1^ assigned to deformation vibration of CNO, CCO, and CCN groups, and at 545 cm^−1^ assigned to stretching vibration of CO and CCl groups and deformation vibration of CCC groups [24,25]. The IR spectrum of KOL shows bands at 3447 cm^−1^ assigned to stretching vibration of OH groups, 2965, 2933, and 2870 cm^−1^ assigned to symmetric and asymmetric stretching vibration of CH groups, 1740 and 1658 cm^−1^ assigned to the stretching vibration of C=O groups from vinyl acetate and pyrrolidone ring, respectively; and also at bands at 1375 cm^−1^ assigned to the stretching vibration of COO groups, at 1236 cm^−1^ assigned to the stretching vibration of CC groups and deformation vibration of CCH and CNH groups, at 1119 cm^−1^ assigned to stretching vibration of CN and CO groups and deformation vibration of CCH and CCC groups, at 1024 cm^−1^ assigned to stretching vibration of CC and CCl groups and deformation vibration of CCH groups, and at 944 cm^−1^ assigned to the stretching vibration of CN groups [26].

The stretching vibrations of OH bonds of chitosan were evidenced at 3444 cm^−1^, and symmetric and asymmetric stretching vibrations of CH groups were observed at 2960, 2923, 2886, and 2866 cm^−1^, respectively. The absorption bands at 1651 cm^−1^, 1597 cm^−1^, 1424 cm^−1^, and 1381 cm^−1^ are assigned to the stretching vibration of C=O groups of amide I, deformation vibrations of the NH (N-acetylated residues, amide II band), deformation vibration of CH_2_, CH_3_ groups and deformation vibration of OH groups. The band at 1258 cm^−1^ is assigned to the stretching vibration of NH, COC, and COH groups, and the bands at 1158 cm^−1^ and 1079 cm^−1^ are assigned to the stretching vibration of CO and COC groups [27].

The FT-IR spectrum of the Aerosil (A) showed specific bands at 1106 cm^–1^ assigned to asymmetric stretching vibrations of the Si-O-Si bonds of silica oxide and a small band at 806 cm^–1^ assigned to symmetric deformation of the Si-O-Si groups.

In the FT-IR spectrum of magnesium stearate (ST), specific bands assigned to the stretching vibration of CH groups appear at 2921 and 2852 cm^−1^, while the bands allocated to the symmetric and asymmetric stretching vibration of (COO-) groups are identified at 1576 and 1456 cm^−1^, respectively. The band at about 3436 cm^−1^ is due to stretching vibrations of the associated water molecules.

The FT-IR spectrum of Avicel (AV) presents characteristic bands of cellulose at 3413, 3346, 1434, and 1324 cm^−1^ assigned to stretching and in-plane deformation vibration of OH groups, at 2901, 1371, and 1276 cm^−1^ assigned to stretching and deformation vibration of CH groups, at 1643 cm^−1^ assigned to stretching vibration of absorbed OH and conjugated CO groups, at 1243, 1163, 1113, 1061, and 1026 cm^−1^ assigned to stretching vibration of CO groups, and at 901 cm^−1^ assigned to the β-glucosidic linkage between the sugar units [28].

To identify if there are interactions between the components of the mixtures, the experimental FT-IR spectra of formulations were compared with their calculated ones (using the additivity low [27] (Figure 7).

In the case of a physical mixture, there should be little or no interaction between the components. The intensity of the spectral bands showed a linear dependence with the sample concentration in the blend. All the characteristic bands of CLZ were present in the spectra of the formulations, and their intensity varied according to their content. The similarity between experimental and calculated spectra indicates that there was no interference of the functional groups, no sign of any polymorphic changes, and the chemical integrity of the drug was not affected.

#### 2.1.2. X-ray Diffraction Analysis

The diffractogram of chlorzoxazone displayed intense signals at 12.98, 13.86, 15.57, 17.83, 19.93, 21.05, 25.22, 25.79, 27.51, and 32.07 (2θ degrees) and a degree of crystallinity of about 80% KOL and CHT have a background pattern with two extensive signals at 13.04 and 21.95 (2θ) and 9.32 and 20.01 (2θ), respectively. The degree of crystallinity was 21.7% for KOL and about 37.2% for CHT. In the case of ST, very intense signals appeared at 1.79, 3.58, 5.36, and 21.46 (2θ degrees), and the degree of crystallinity was about 49.5%. The AV diffractogram presents the specific signals assigned to cellulose, namely: 15.1 degrees assigned to the (101) plane, 16.4 degrees assigned to the (101) plane, and 22.7 degrees assigned to the (200) plane of cellulose I [28,29].

The physical mixtures showed characteristic signals of the pure components at identical angles, proving that no interactions occurred during mixing (Figure 8). The signals of CLZ were slightly low in intensity in the physical mixture due to a lower drug concentration. This observation supported the absence of any chemical interactions between the drug and the other components of the mixture.

It can be concluded that FT-IR spectroscopy and XRD analyses revealed no interactions between active principle and excipients. KOL, CHT, A, ST, and AV used in this study do not interact with the CLZ and may be valuable excipients in pharmaceutical formulations.

#### 2.1.3. Thermal Characterization

The data of Figure 9a,b compare the differential thermogravimetric curves (DTG) curves for CLZ, CHT, AV, KOL, ST, and the formulations (F1b, F2b, and F3b). The recorded TG, DTG, and differential thermal analysis (DTA) curves and their interpretation with Mettler Toledo’s STAR software 2.7.11b revealed the main thermal characteristics given in Table 4 and Table 5.

According to the presented data in Table 4, except CLZ, all samples showed a first stage of moisture removal ranging from 2.5 wt% to 12.3 wt%.

The AV sample shows a mass loss of approximately 80% within the 292–378 °C temperature range, specific to cellulose decomposition in an inert atmosphere. The thermogravimetric curves recorded in the present research are similar to those reported by other researchers for AV samples recorded under similar conditions [30,31].

Once the 12.21 wt% moisture content was removed, the thermal decomposition of CHT occurred in two steps, with the most significant mass percentage loss occurring within the 277–302 °C temperature range. A mass loss of about 43 wt% in the second stage of decomposition, due to the decomposition of chitosan polymer chains by deacetylation and cleavage of glycosidic bonds, was also reported by Rahman et al. [32] for chitosan samples that were analyzed under similar conditions. The last-step mass loss for CHT may be linked to the thermal decomposition of the pyranose ring and of the residual carbon.

CLZ decomposes in a nitrogen atmosphere in a single step, resulting in a residue amount of 4.40 wt%. Roy and Ghosh [33] also analyzed the decomposition of CLZ in a nitrogen atmosphere at the rate of 10°C/min within the 30–700 °C temperature range. They identified a single degradation stage within the 210–290 °C range.

The KOL sample contains 3.75 wt% moisture, which is removed up to 80 °C. The most significant mass loss of this sample occurs within the 320–361°C temperature range and is due to the deacetylation of polyvinyl acetate present in KOL [34]. The mass loss recorded within the 416–473 °C temperature range of 32.49 wt%, which may be linked to the production of aromatic hydrocarbons [35], but also to the depolymerization of polyvinylpyrrolidone presented in a smaller amount in KOL [36].

The thermal decomposition of ST occurs within the 330–410 °C temperature range, also identified by other researchers by thermogravimetric analysis under similar conditions [37]. This step may be linked to the denitrification process, which consists of the NO_2_ and O_2_ removal and forming MgO [37].

Thermal decomposition occurred in four steps in the case of the F1, F2, and F3 formulations. The first stage consists of moisture removal, which ranged between 0.8 and 1.5 wt%. The second step may be linked to the decomposition of CLZ within the 232–303 °C temperature range and with the temperature at which the decomposition rate reached its peak (T_peak_) of about 281 °C. According to the obtained results, the most significant amount of active substance CLZ was found in the F1 formulation (Figure 9b and Table 5). A slight difference regarding the amount of CLZ was found in the F2 and F3 formulations, and the TG curves were slightly different within the 232–303 °C temperature range, according to the data in Figure 10. According to the data referred to above, the thermal decomposition of the excipients present in the formulations occurs in a single stage within the 300–400 °C temperature range, with T_peak_ = 361 °C for AV, 302 °C for CHT, 343 °C for KOL, and 386 °C for ST.

The F1b, F2b, and F3b formulations have a single peak at 338 °C in the third stage, within the 326–359 °C temperature range, which proves an excellent compatibility of the excipients used [38]. According to the TG curves shown comparatively in Figure 10 and to the main thermogravimetric characteristics shown in Table 5, the last step of decomposition shows more significant differences between the three formulations, which may be linked to various amounts of KOL. This is accounted for by the fact that, according to the data in Table 4, the last stage of thermal decomposition of the KOL excipient occurs within the 416–473 °C temperature range. According to Figure 6b, at temperatures above 500 °C, a small peak can be distinguished for the F2b formulation that could be linked to the last stage mass loss in the case of CHT. This would indicate more of this excipient in the F2b formulation.

#### 2.1.4. DSC Results

DSC curves were recorded under nitrogen at a 10 °C/min heating rate and two heating runs, and one cooling run have been performed. The results obtained for the components of the formulations are compared in Figure 11a–c.

According to the DSC curves shown in Figure 8a for AV, an endothermic peak at 76 °C linked to the desorption of moisture from microcrystalline cellulose and an endothermic peak corresponding to mannitol melting peak at 168 °C were noted [39].

CHT peaks at 81 °C were noted in the DSC curve during the first heating run (Figure 8a), which may be linked to the evaporation of residual water involving an enthalpy variation of 275.69 J/g. These findings are close to those reported by other researchers in the literature [40].

KOL showed three endothermic peaks at 44 °C, 89 °C, and 115 °C, respectively, according to the DSC curve for the first heating run within the 25–250 °C temperature range, as shown in Figure 3a. According to the DSC curve obtained in the second heating run (Figure 8c), this excipient underwent two glass transitions at temperatures of 43 °C and 172 °C, as it is a copolymer of polyvinyl acetate (PVA) and polyvinylpyrrolidone [41].

Similar to other data reported in the literature, CLZ showed a melting peak at 191 °C during the first and second heating runs and a crystallization peak at 181 °C during the cooling run [42].

ST, the most widely used excipient in solid oral pharmaceutical formulations, showed six endothermic peaks on the first heating curve [43]. The first two, at 73 °C and 109 °C, respectively, are specific to dehydration processes [44]. They no longer occur in the second sample heating run within the 25–250 °C temperature range. Only one exothermic peak occurred on the cooling curve of ST at 126 °C.

Figure 12a–c compare the DSC curves recorded for the F1b, F2b, and F3b formulations.

The DSC curves for the first heating stage confirm the presence of the active substance CLZ and its compatibility with the excipients used. Thus, we found that CLZ preserved its crystalline form in the presence of excipients and that the melting peak temperature decreased from 191 °C to 163 °C, 164 °C, and 172 °C, respectively. The DSC curves recorded in the cooling run and shown in Figure 9b indicate the presence of peaks corresponding to the CLZ crystallization but at lower temperatures than in the case of CLZ in the absence of excipients (Figure 9b). Enthalpy also varies in the crystallization process, especially for F1b formulation (31.12 J/g) than for formulations F2b (24.46 J/g) and F3b (23.08 J/g). These results confirmed data reported from the thermogravimetric curve analysis, that the F1b formulation has the highest amount of active substance CLZ. In contrast, the F2b and F3b formulations have little difference in the same amount of CLZ. In the characteristic curves of the second heating run (Figure 9c) melting peaks and glass transition temperatures are recorded at approximately 30 °C and 112 °C that may be linked to the presence of KOL.

### 2.2. In Vitro Dissolution Studies

The results obtained during the in vitro dissolution test reveal the prolonged release of CLZ from the studied formulations when compared to the release of CLZ from an industrial product [45]. On the other hand, these results also highlighted the central role of the KOL exerted on matrix tablet release characteristics. The released amount of CLZ varied with the percentage of KOL for all studied formulations (Figure 10). A particular behavior was observed for F2b containing 30 (w/w) % KOL, released 26.39 (w/w) % of CLZ during the first two hours of the dissolution test in simulated gastric fluid. Moreover, that the formulation released 87.05 (w/w) % CLZ of the 12 h and 98.55 (w/w)% at the end of the 36 h of testing. It was also found that by the increasing the CHT concentration in the formulation decreased the CLZ release rate from the matrix tablets. Thus, in the F1c, F2c, and F3c formulations containing 40 (w/w) % KOL and 7 (w/w) % CHT, at the end of the dissolution test, the amount of CLZ is the smallest. The F1b containing 5% CHT released 95.76% of CLZ, F2b with 5% CHT released 98.55% CLZ, and F3b with 5% CHT generated only 85.15% CLZ (Figure 13).

The dissolution profiles have been compared considering CLZ content from the formulations studied by calculating the similarity factor f2 and the difference factor f1, considering formulations F1a, F1b, and F1c as reference formulations, and formulation F2b-F3b as test formulations. The results obtained (Table 6) confirmed that both CHT and KOL influenced the release characteristics of the matrix tablets, and virtually each studied formulation had its particular release kinetics.

Although the similarities between the three release profiles of CLZ were confirmed by the value of the similarity factor (f2 = 57.9562, f2 = 51.5645), the difference factor had a value greater than 15 (f1 = 33.7543, f1 = 35.8516), which corresponded to a difference of more than 10% between the F1a, F1b, and F1c analyzed profiles.

### 2.3. Drug Release Kinetics

The CLZ release study was performed in two steps, modifying the pH of release media to cover the physiological pH range, from 1.2 pH buffer simulating the gastric fluid to 6.8 pH buffer mimicking the intestinal fluid (Figure 14). A gradual release process that extends over several hours (up to 36 h) was recorded, which could be attributed to a slow CLZ diffusion through the prepared formulations in both pH buffer solutions. In the 12–36 h interval, the release profiles are close to reaching the equilibrium. At the end of the release process, the total released amount (Qmax, %) of CLZ from the formulations ranged between 86.1% and 98.8% (Table 7), where the F2b formulation registered the highest drug released percentage.

The results of the in vitro release tests have been evaluated from the kinetic perspective (fitted with the Korsmeyer–Peppas equation), and the kinetic parameters (*n* and *k*) were calculated to establish the mechanism and rate of drug release (Table 8). The equation was applied to the first (0–2 h) and second (2–36 h) steps of drug release.

Evaluating the first step of the kinetic release profile, the value of release exponent n for all the formulations was under 0.5 (n = 0.077–0.196), which is associated with a pseudo-Fickian mechanism. A different mechanism of the drug release, namely an abnormal or non-Fickian diffusion mechanism, was characteristic of the second step of the kinetic release profile, where the *n* value ranged between 0.5 and 1 (n = 0.58–0.70).

It can be observed that the values of release rate constant *k* are well correlated with the release profiles presented in Figure 11. The highest values, 0.197 h^—n^ in the first step and 0.17 h^—n^ in the second step, were registered for the F2b sample, data that suggest a faster drug release.

In Table 8, values of R^2^ range between 0.983 and 0.991. For R^2^, the value should be as close as possible to 1 to demonstrate a yield as good as possible for a formulation.

In a study by Raval and Co. [46], the solubility of pure CLZ drug was higher in pH 6.8 buffer solutions, whereas it was low in pH 1.2 buffer solutions. However, in our study, the release rate constant, k, is different for both steps of the release profiles, remaining dependent on the pH of the buffer solution, meaning that after the incorporation of CLZ in the three formulations, the release by diffusion is influenced by the pH sensibility of the drug solubility. It can be concluded that the incorporated formulations with CLZ are suitable for use as oral delivery systems that provide a controlled and prolonged release over 36 h.

## 3. Materials and Methods

### 3.1. Materials

Chlorzaxazone (Orchid Chemicals Ltd., Chennai, India), Kollidon^®^SR (BASF, Ludwigshafen, Germany), chitosan (practical grade, BASF, Germania), Avicel^®^PH (Chemtrec, Falls Church, VA, USA), Aerosil^®^200 (Degussa, Frankfurt, Germania), and magnesium stearate (Union Derivan S.A., Barcelona, Spain).

In this study, a CHT with a degree of deacetylation from 51% to 65% has been used because only such kind increases the absorption of active substances hydrophores with high molecular weight [17,18]. It can be considered as a chitin/chitosan copolymer with formulae given in Figure 3.

Other materials used wrre: Avicel^®^ PH (AV): microcrystalline cellulose. Aerosil^®^ (A): hydrophilic fumed silica with a specific surface area of 200 m² g^−1^ and magnesium stearate (ST) was also used. All used compounds accomplish the quality requirements according to the laws of force. These excipients facilitate the application of the direct compression method to obtain optimal hydrophobic substances (CLZ) dispersibility in the powdered mixtures and the association of hydrophilic chitosan.

### 3.2. Characterization of the Formulations of CLZ Matrix Tablets

The formulation characterization according to standard procedures, includes determining flow time, compressibility parameters of powder mixtures, coefficient of friction, angle of repose, Hausner ratio, and Carr index.

The analysis of the data obtained in this stage is essential in carrying out a proper compression process because the flow properties of a mixture can be affected by shear forces, surface tension, electrostatic forces, van der Wals forces, and mechanical forces resulting from the interaction of the particles [18]. The Hausner ratio and the Carr index are indicators of the compressibility characteristics of the powders. Analyzing and improving these parameters in the formulating stages allow us to obtain tablets with optimal mechanical resistance without the tendency to pickle. Both CHT and KOL (as raw materials) are characterized by optimal values of these parameters, recommending them as excipients for direct compression [47,48].

Determination of flow and compressibility parameters of powder mixtures was carried out on powdered mixtures for the nine proposed formulations, using 50, 33, and 25 (w/w %) CLZ, 20–40 (w/w %) KOL, and 3–7 (w/w %) CHT. The auxiliary substances were formulated in constant concentrations according to Table 1 (see above) [45,49].

#### Flow Parameters of the Powder Mixtures

To evaluate the flow and compressibility parameters of the powder mixtures, the following indicators were determined:

Flow time (g/s) was determined by recording the time required for 50 g of powder to flow through a funnel with a 10 mm hole.

Coefficient of friction (tg α) was determined by the dynamic method, using the Equation (1):
tg α = h/r (1)
where h = height and r = the radius of the powder cone.

Angle of repose (α) was determined using the dynamic method.

The Hausner ratio, which is calculated by relating the bulk density of a powder to the value of the compaction density, is a parameter related to the cohesion and adhesion forces that exist between the particles [50].

Hausner ratio (R_H_) was determined by measuring the density before (ρi) and after compaction (ρc), according to the Equation (2):
R_H_ = ρc/ρi (2)

The Carr compressibility index is a parameter indicating the ability of some powders to form a homogeneous mixture.

Carr index (Ic) was determined using the density measurements undertaken for the Hausner ratio [20], according to Equation (3):
Ic = (ρc − ρi)/ρc × 100(3)

### 3.3. Preparation of CLZ Matrix Tablets

Matrix tablets were prepared using the power mixtures for the nine proposed formulations by direct compression with the Korsh EK0 compression machine (10 mm ponson diameter, 8–10 kN compression force) [51].

The mixture of powders corresponding to the formulation of matrix tablets with CLZ were mixed in an Erweka AR 403 mixer (Erweka GmbH, Heusenstamm, Germany) with a rotation speed of 400 rpm for 5 min, after which they were sieved using an EM-8 electromagnetic sieve (Erweka GmbH, Heusenstamm, Germany), then they were subjected to compression directly to the Korsch EK0 compression machine (Korsch AG, Berlin, Germany). Table 1 presents the compositions of the nine oral formulations, where besides KOL and CHT, other specific auxiliary excipients were included [52]. In 100 (w/w) % powder mixture 50 w/w %, 33 w/w %, or 25 w/w % represent CLZ. The proportion of KOL (20-40 w/w %) and CHT 3–5 (w/w %) was varied, the ST and AE constant values and the AV completes at 100 w/w %.

These proportions were taken into account when preparing the tablets, but the average weight per tablet is 500 mg. A matrix tablet can contain 250 mg CLZ (F1), 165 mg CLZ (F2), or 125mg CLZ (F3).

The matrix tablets with an average diameter of 10 mm and thickness of 3.0 mm have been obtained.

The effects of KOL and CHT were studied individually, and their combination at 1:1, 1:2, and 1:3 CLZ/excipients ratios. By this strategy has been evaluated the influence of therapeutic system type on the oral disponibility of CLZ.

### 3.4. Evaluation of Matrix Tablets

#### 3.4.1. Pharmaco-Technical Parameter of Matrix Tablets

The quality of the matrix tablets was assessed by determining the pharmaco-chemical characteristics of hydrophilic matrix tablets with the modified release [51,53]: weight uniformity, drug content uniformity, friability, mechanical strength, thickness, and in vitro dissolution studies.

Weight uniformity was determined according to the 8th European Pharmacopoeia [22] by weighing 20 tablets on the Radwag WPE 60 electronic balance [54]. Twenty tablets were randomly selected from each formulation and weighed individually. The individual weights were compared with the mean weight, and the standard deviation (SD) was calculated [54].

Dose uniformity was evaluated by quantitatively determining CLZ in tablets using an HPLC method. Ten were collected and powdered in a mortar for each formulation of prepared tablets. 2.0–5.0 mg power was taken in a volumetric flask to add 10 mL of methanol, and the solution was sonicated for 10 min for complete solubilization of CLZ. The solution was filtered, 1 mL filtrate was diluted with 10 mL phosphate buffer pH 7.4, resulting in 0.020–0.025 mg CLZ mL^−1^, and analyzed by the HPLC method previously described [13] with slight modification.

HPLC was performed using a chromatograph of Thermo Fisher Surveyor type (Thermo Fisher, San Jose, CA, USA) equipped with a UV-VIS detector with multiple diode array detectors and a Thermo Fisher-Hypersil Betasil C18 150 mm × 4.6 mm column (Thermo Fisher, San Jose, CA, USA), the particle size dimension was of 5 µm. The column temperature was kept constant at 45 ± 0.2 °C. As the mobile phase, a mixture of 0.05 M dihydrogen phosphate and methanol in the 50:50 *v*/*v* ratio was used at a flow rate of 1.5 mL min^−1^. The injection volume for each determination was 20 μL. CLZ was detected by the UV spectrum at its characteristic wavelength of 280.4 nm.

Thickness, diameter, and mechanical strength.

The evaluation of the pharmaco-technical characteristics (diameter, thickness, and average mass) of the tablets was performed according to the 8th European Pharmacopoeia [22]. The weighing was carried out by means of an electronic balance Redwag WPE 60 on 29 tablets for mass and dose uniformity determination.

Mechanical resistance was performed on ten tablets on a Schleuniger Pharmatron tablet hardness tester 8M (Sotax AG, Aesch, Switzerland).

From thickness variation, ten tablets from each formulation were taken randomly, and their thickness was measured using a micrometer (Micro-Epsilon Messtechnic GMBH & CO.KG, Ortenburg, Germany). The mean thickness and SD were calculated.

Friability was determined on 20 tablets on Schleuniger Pharmatron FTII friability tester (Sotax AG, Aesch, Switzerland) at 100 rpm for 4 min. The tablets were reweighed, and the percent friability was then calculated according to the following Equation (4):Friability (%) = (w_1_ − w_2_)/w_1_ × 100(4)
where w_1_ is the initial weight of the tablets, and w_2_ is the final weight after tumbling.

Hydration capacity or swelling degree was determined by using a dissolution test station type II, Hanson SR 8 Plus Series (Hanson Research Co., Chatsworth, CA, USA). Matrix tablets have been introduced in 1000 mL distilled water at 37 ± 2 °C at 60 rpm. At the predetermined time intervals (1 h), samples prevailed from the hydration medium, and the excess water on the surface was removed by wiping with filter paper and then weighed. The swelling degree [55] was evaluated using relation 5:SD = [(W_t_ − W_o_)/W_o_] × 100(5)
where SD = swelling degree; W_t_ = mass of the sample at time t; W_o_ = dry mass of the tablet.

The three formulations, F1b, F2b, and F3b, showed optimum pharmaco-technical properties, and they had the greatest potential to be used in oral pharmaceutical products for the controlled delivery of CLZ.

#### 3.4.2. Drug-Excipients Compatibility Study

##### Fourier Transform Infrared Spectroscopy

FT-IR spectroscopic technique showed that drug–excipients chemical interaction may occur in the formulation due to their intimate contact. The spectrum of CLZ physical mixture and the formulation were obtained by KBr pellet method. The FT-IR spectra were recorded in the 4000–500 cm^−1^ region with 4 cm^−1^ spectral resolution on a Bruker ALPHA FT-IR spectrometer. The concentration of the pellets was 3 mg sample/300 mg KBr for all studied formulations.

##### X-ray Diffraction

The diffractograms were recorded on a Diffractometer D8 ADVANCE (Bruker AXS, Germany), using the CuKα radiation. The working conditions were 40 kV, 30 mA, a 2 s/step, and 0.02 degrees/step. All diffractograms were recorded in the 10–40 2θ degrees range at room temperature.

##### Thermal Characterization

Thermogravimetric (TG), derivative thermogravimetric (DTG), and differential thermal analysis (DTA) curves were recorded with a Mettler Toledo 851^e^ device under a nitrogen atmosphere at the heating rate of 10 °C/min within the 25–700 °C temperature range. Samples weighing 2.2 to 3.7 mg were used, and the nitrogen flow rate was 20 mL/min. We checked the reproducibility of the data collected by recording three tests for each sample under the conditions described above. The uncertainty was found to be less than 1%.

Differential scanning calorimetry (DSC) curves were recorded with a Mettler Toledo DSC1 device. The tests were performed at a heating rate of 10°C/min in an inert atmosphere (nitrogen), with two heating runs and one cooling run. −20–210 °C was the temperature range in which scans were performed for the formulations (F1b, F2b, and F3b), 25–210 °C for CLZ, 25–230 °C for CHT, and avicel (AV), and 25–250 °C for magnesium stearate (ST) and KOL, respectively. The scanned samples weighed between 2 and 4.1 mg.

The TG, DTG, DTA, and DSC curves were processed using Mettler Toledo’s STAR software to obtain the main thermal characteristics.

#### 3.4.3. In Vitro Dissolution Studies

The CLZ dissolution profiles from the matrix tablets have been studied into two dissolution media with different pH, namely: HCl 0.1 N solution with pH = 1.2 (simulating media for gastric fluids) and phosphate buffer solution with pH = 6.8 (simulating media for intestinal fluids). The experiments were carried out by means of a dissolution test station type II, Hanson SR 8 Plus Series (Hanson Research Co., Chatsworth, CA, USA) provided with two blades. The experiments were performed according to the requirements described in Romanian, European, and United States Pharmacopoeia (USP, with specifications for liquid and solid pharmaceutical formulations).

The dissolution medium was a pH 1.2 solution for the first 2 h, and then, pH 6.8 solution (phosphate buffer) for the next 34 h. SR 8 Plus Series blades apparatus type II set at 37 ± 0.5 °C and 50 rpm; the sampling interval was set every hour during the 36 h test (5 mL of sample were replaced with the same volume of medium). The quantitative determination of CLZ was performed using an HPLC method. The results were interpreted and statistically analyzed using Matlab 7.9.

According to pharmaco-technical specifications for the preparation of modified release tablets, the release profiles of the active substance from such type of tablets must be analyzed by determining the dissolution test for solid pharmaceutical forms, difference factor *f1* and the similarity factor *f2,* and the correlation coefficient R^2^ between two or more formulations [52,56].

The difference factor f1, and the similarity factor f2, have been calculated according to the following equations:(6)$$f_{1}=\left\{\frac{\sum_{t=1}^{n}\left|R_{t}-T_{t}\right|}{\sum_{t=1}^{n}R_{t}}\right\}\times 100$$
and
(7)$$f_{2}=50{\;\log}_{10}\left\{{\left[1+\frac{1}{n}\sum_{t=1}^{n}{\left(R_{t}-T_{t}\right)}^{2}\right]}^{-0.5}\times 100\right\}$$
where n is the number of points for specimen collection, R_t_ is the amount of dissolved active substance from the reference formulation at the t moment, T_t_ = the amount of dissolved active substance of the studied formulation at the t moment, and log_10_X = 10 base logarithm of X.

#### 3.4.4. Drug Release Kinetics

The drug release data up to 60% of the total drug released were fitted using the Korsmeyer–Peppas equation (Equation (1)) to establish the release mechanism [57]:(8)$$\frac{M_{t}}{M_{\infty }}=kt^{n}$$
where *M_t_/M_∞_* = fraction of the drug released at time *t*, *M_t_* = absolute cumulative amount of drug released at time *t, M_∞_* = maximum amount released in the experimental conditions used, at the plateau of the release curves, *k* = release constant and *n* = release exponent, which is indicative of the release mechanism.

In the above equation, a value of n = 0.5 indicates a Fickian diffusion mechanism of the drug from the samples, while a value of 0.5 < n < 1 indicates anomalous or non-Fickian behavior. When n = 1, a case II transport mechanism is involved with zero order kinetics, while n > 1 indicates a special case II transport mechanism [58].

To analyze the mechanism of drug release from matrix tablets, the results of in vitro release data were plotted using various kinetic models like zero order, first order, Higuchi, and Krosmeyer–Peppas models.

The evaluation of release profile kinetics of CLZ from matrix tablets based on KOL and CHT was carried out using analysis by fitting into four representative mathematical models, which are basic elements for understanding the mechanisms underlying the release of active substances from the studied formulations.

The data fitting was carried out by linear or non-linear regression using Matlab 7.1. The correlation coefficient R^2^ was the criteria for selecting the model that most faithfully depicted the release profile of each studied formulation. A prediction as good as possible of the model requires R^2^ to be as close to 1 [59].
(9)R2=1−∑i−1nyi−y^i2∑i=1nyi−y¯2
where: y_i_ = experimental data, y^i = values approximated by model, and $\bar{y}$ = average of experimental data.

## 4. Conclusions

In the present study, nine formulations of matrix tablets with CLZ were obtained and characterized to develop CLZ tablets with the modified release. KOL and CHT were used as matrix-forming agents, and their influence on the flow and the compressibility properties of the powders were analyzed as well as their effect on the pharmaco-chemical characteristics of the matrix tablets. The formulations of matrix tablets F1a,b, F2a,b, and F3a,b obtained through direct compression, containing 20–30 (w/w) % KOL, exhibited optimal flow properties and compressibility. In comparison, formulations with more than 40 w/w % KOL had a poor flow or even lacking. The pharmaco-chemical characteristics of the tablets, defined both by the working conditions and the flowing and compressibility characteristics of the powders, were directly influenced by the matrix-forming polymers used to obtain hydrophilic matrix tablets with modified release. The mechanical strength of the tablets varied inversely to the KOL concentration in the formulation. Concentrations of KOL greater than 30 (w/w) % lead to improper matrices regarding friability and mechanical strength. CHT did not significantly influence the pharmaco-chemical properties of the formulations studied (mechanical strength, friability, diameter, thickness, mass uniformity). However, that polymer influenced the matrix release characteristics, as increasing CHT concentration decreased the CLZ release rate.

The three formulations F1b, F2b, and F3b showed optimum pharmaco-technical properties, and they had the most significant potential to be used in oral pharmaceutical products for the controlled delivery of CLZ.

The FT-IR spectroscopy and XRD analysis revealed no interactions between active principle and excipients. KOL, CHT, A, ST, and AV used in this study do not interact with the CLZ and may be valuable excipients in pharmaceutical formulations.

The DSC curves for the first heating run confirm the presence of the active substance CLZ and its good compatibility with the excipients used. Thus, we found that CLZ preserved its crystalline form in the presence of excipients and that the melting peak temperature decreased from 191 °C to 163 °C, 164 °C, and 172 °C, respectively.

The in vitro dissolution test showed good results for the CLZ tablets, and they proved to be suitable for the prolonged and controlled release of CLZ. After 2 h in simulated gastric fluid with pH = 1.2, the CLZ release was 20–26%; it was approximately 54–70% in the simulated intestinal fluid with pH = 6.8 after 7 h and 85–98% after 36 h.

The values of the difference factor f1, the similarity factor f2, and the correlation coefficient R^2^ showed that the three formulations studied are different in terms of the release profile.

The in vitro kinetic study revealed a complex mechanism of release occurring in two steps through a pseudo-Fickian mechanism. The first step of the kinetic release profile and the non-Fickian diffusion mechanism were characteristic of the second step. The incorporated formulations with CLZ are suitable for use as oral delivery systems that provide a controlled and prolonged release over 36 h. In vitro dissolution tests revealed that the F2b sample behavior is different, which suggests a faster drug release.

In conclusion, the results confirmed that CLZ can be formulated as hydrophilic matrix tablets based on KOL and CHT, with up to 30 (w/w) % KOL and 5 (w/w) % CHT. The present matrix tablets are aimed at improving patient compliance, reducing the number of administered doses, and the number of administered doses being reduced.

Formulations F1b, F2b, and F3b will be studied in vivo to determine the oral bioavailability of CLZ.

## Figures and Tables

**Figure 1 ijms-25-05137-f001:**
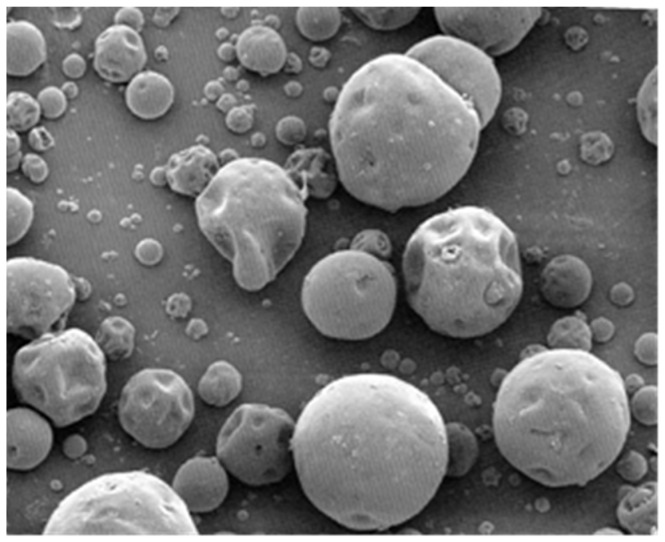
The SEM image of the KOL particles according to manufacturer specifications.

**Figure 2 ijms-25-05137-f002:**
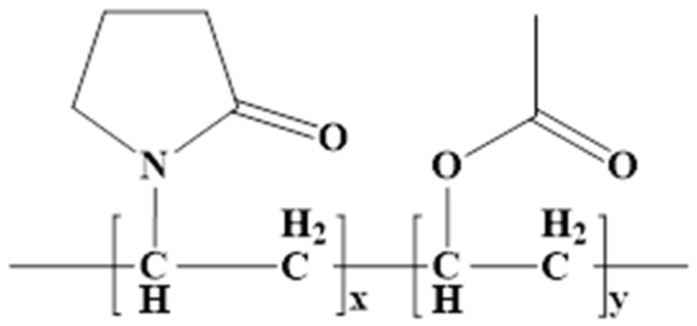
General molecular formula of KOL (polyvinyl pyrrolidone radical x = 450; polyvinyl acetate radical y = 5200).

**Figure 3 ijms-25-05137-f003:**
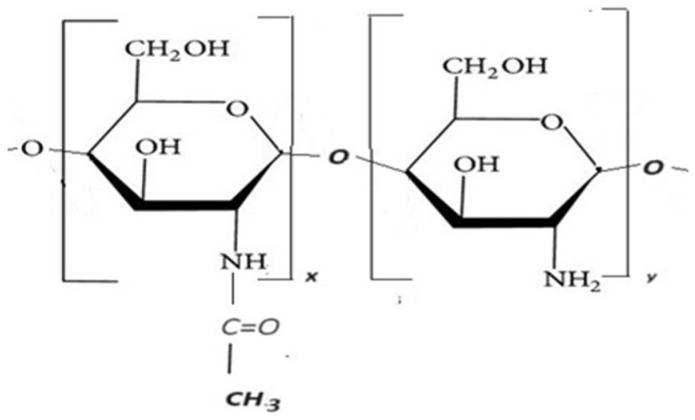
Chemical structure of CHT.

**Figure 4 ijms-25-05137-f004:**
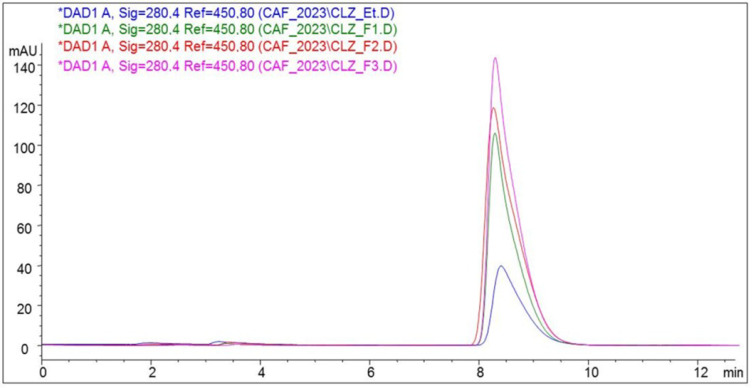
HPLC chromatograms of the CLZ and three formulations (F1, F2, and F3).

**Figure 5 ijms-25-05137-f005:**
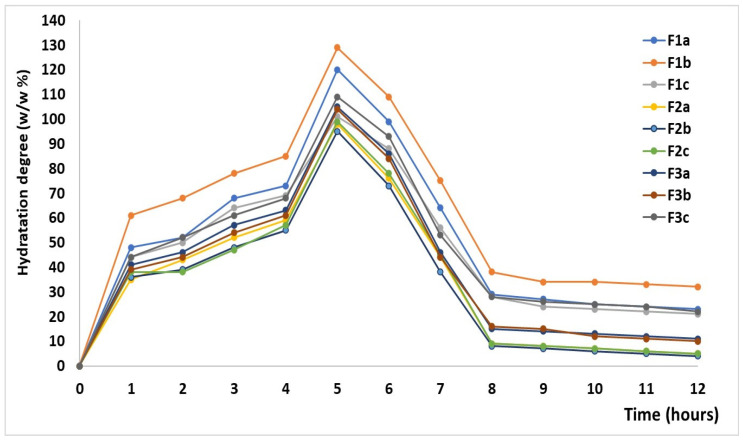
Variation of hydration degree of the matrix tablets.

**Figure 6 ijms-25-05137-f006:**
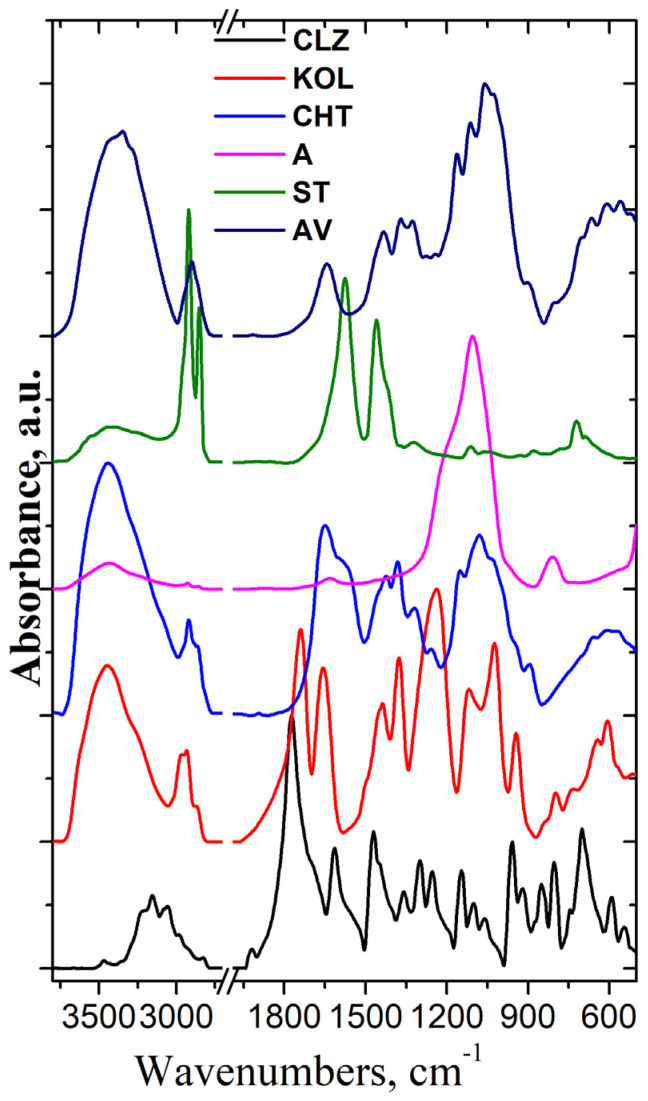
FTIR spectra of the components of the formulations.

**Figure 7 ijms-25-05137-f007:**
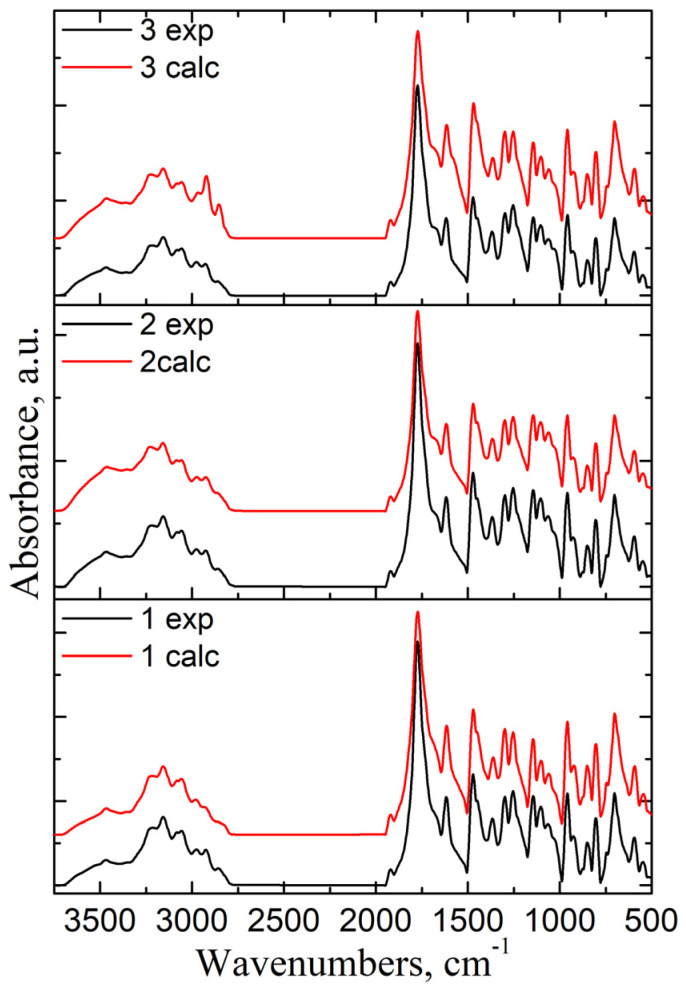
Comparison between experimental and calculated IR spectra of some studied formulations F1b, F2b, and F3b.

**Figure 8 ijms-25-05137-f008:**
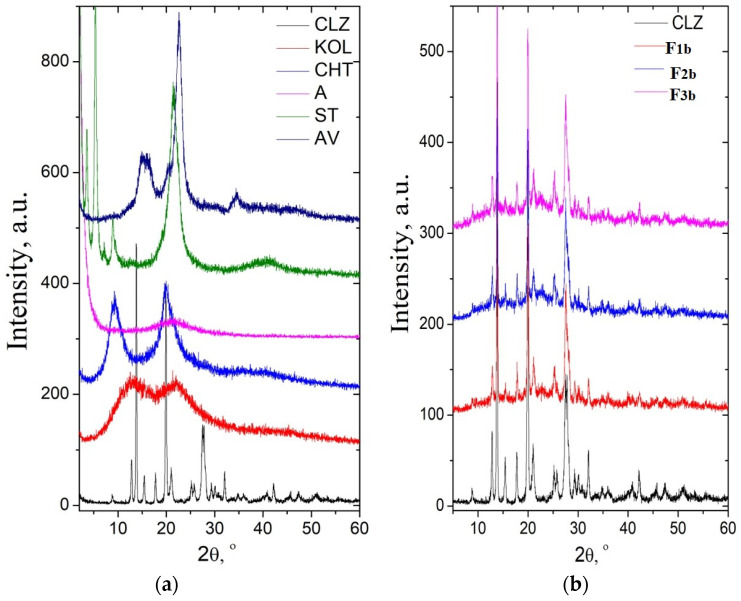
XRD diffractograms of pure components (**a**) and the mixtures with composition of the F1b, F2b, and F3b formulations (**b**).

**Figure 9 ijms-25-05137-f009:**
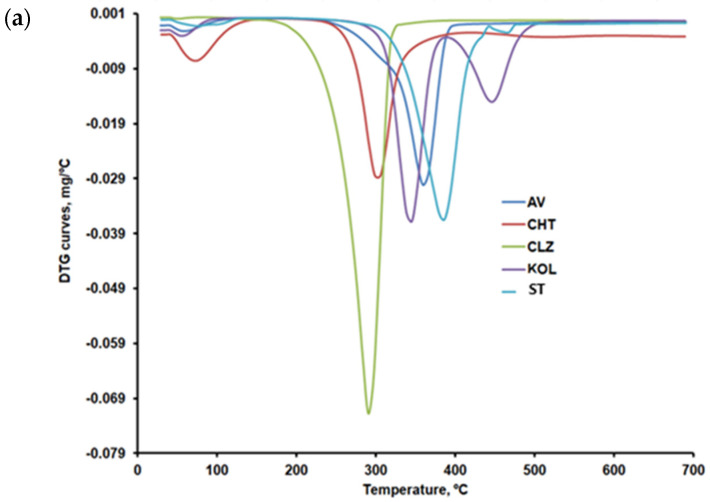
(**a**) DTG curves for CLZ, CHT, AV, KOL, and ST. (**b**) DTG curves for formulations F1b, F2b, and F3b.

**Figure 10 ijms-25-05137-f010:**
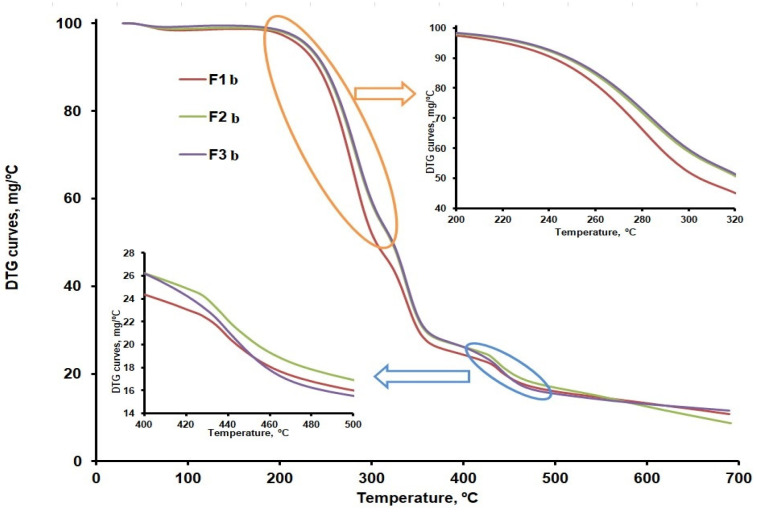
TG curves for the F1b, F2b, and F3b formulations.

**Figure 11 ijms-25-05137-f011:**
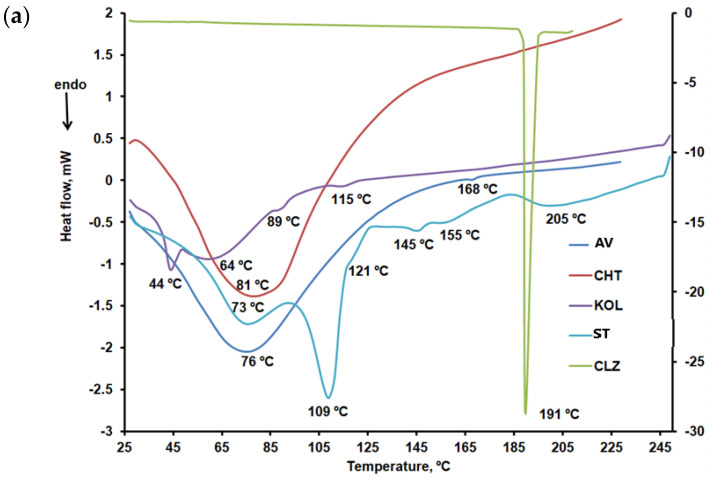
(**a**) DSC curves: CLZ within the 25–210 °C temperature range, AV and CHT within the 25–230 °C temperature range, and KOL and ST within the 25–50 °C temperature range (first heating run). (**b**) DSC curves: CLZ within the 25–210 °C temperature range, AV and CHT within the 25–230 °C temperature range, and KOL and ST within the 25–250 °C temperature range (cooling run). (**c**) DSC curves: CLZ within the 25–210 °C temperature range, AV and CHT within the 25–230 °C temperature range, and KOL and ST within the 25–250 °C temperature range (second heating run).

**Figure 12 ijms-25-05137-f012:**
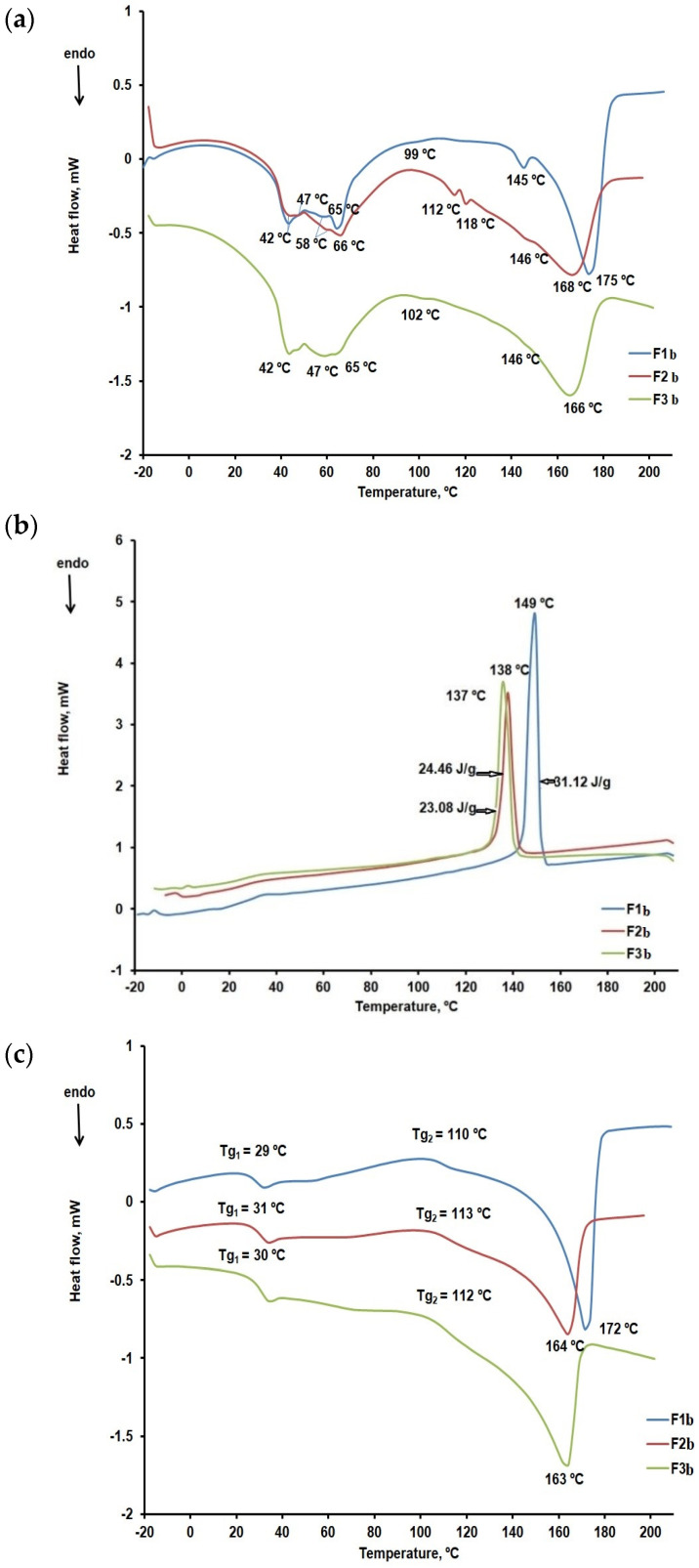
(**a**) DSC curves of the formulations within the 25–210 °C temperature range (first heating run). (**b**) DSC curves of the formulations within the 25–210 °C temperature range (cooling run). (**c**) DSC curves of the formulations within the 25–210 °C temperature range (second heating run).

**Figure 13 ijms-25-05137-f013:**
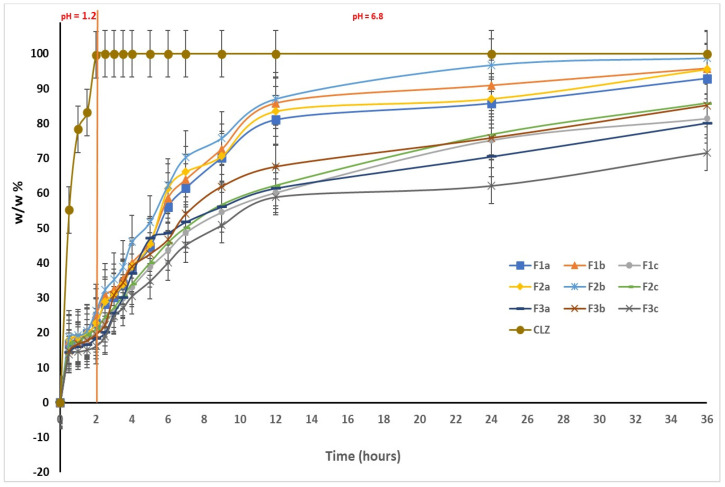
In vitro dissolution profiles of CLZ in F1a,b,c, F2a,b,c, and F3a,b,c.

**Figure 14 ijms-25-05137-f014:**
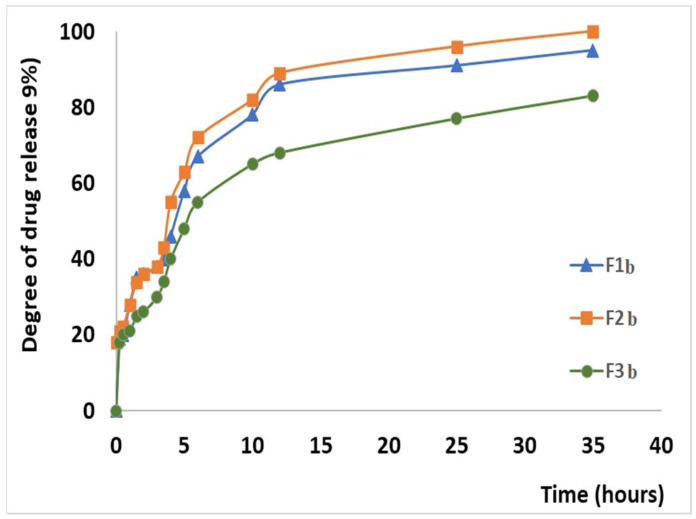
In vitro dissolution profile of chlorzoxazone release mimicking the physiological pathway.

**Table 1 ijms-25-05137-t001:** Formulation of CLZ matrix tablets.

Component (w/w %)	Formulation								
	F1a	F1b	F1c	F2a	F2b	F2c	F3a	F3b	F3c
	1:1			1:2			1:3		
CLZ	50	50	50	33	33	33	25	25	25
KOL	20	30	40	20	30	40	20	30	40
CHT	3	5	7	3	5	7	3	5	7
Aerosil	1	1	1	1	1	1	1	1	1
Mg stearat	0.5	0.5	0.5	0.5	0.5	0.5	0.5	0.5	0.5
Avicel	up to 100								

**Table 2 ijms-25-05137-t002:** Flow parameters of the powdered mixtures.

	Formulations	F1a	F1b	F1c	F2a	F2b	F2c	F3a	F3b	F3c
Parameter										
Flow time (T) (g/s)		0.1254 ± 0.12	0.1368 ± 0.06	0.1525 ± 0.09	0.1133 ± 0.06	0.1233 ± 0.111	0.1167 ± 0.625	0.1211 ± 0.05	0.1147 ± 0.07	0.1471 ± 0.68
Friction coefficient (tg α)		0.4312	0.4602	0.5511	0.4101	0.4368	0.5266	0.4255	0.4073	0.5032
Angle of repose α (degrees)		24.5 ± 0.135	23 ± 0.112	25 ± 0.567	33 ± 0.105	34.5 ± 0.857	35 ± 0.156	35.5 ± 0.525	40.5 ± 0.452	40.6 ± 0.65
Hausner ratio		1.1653 ± 0.05	1.2511 ± 0.15	1.3155 ± 0.16	1.2613 ± 0.19	1.2501 ± 0.105	1.3160 ± 0.07	1.3475 ± 0.08	1.2595 ± 0.07	1.3955 ± 0.09
Carr index (%)		23.113 ± 0.061	24.035 ± 0.155	25.815 ± 0.162	25.011 ± 0.087	25.5166 ± 0.098	26.8751 ± 0.053	25.470 ± 0.075	24.254 ± 0.069	35 ± 0.093

**Table 3 ijms-25-05137-t003:** Pharmaco-technical parameter values of matrix tablet formulations.

	Formulations	F1a	F1b	F1c	F2a	F2b	F2c	F3a	F3b	F3c
Parameter										
Diameter (mm)		10.075 ± 0.015	10.061 ± 0.005	10.101 ± 0.005	10.076 ± 0.015	10.075 ± 0.015	10.083 ± 0.014	10.064 ± 0.011	10.062 ± 0.006	10.058 ± 0.014
Thickness (mm)		4.654 ± 0.015	4.835 ± 0.025	4.843 ± 0.024	4.943 ± 0.191	5.095 ± 0.083	4.743 ± 0.045	4.572 ± 0.056	4.558 ± 0.144	4.523 ± 0.043
Average mass (g)		0.483 ± 1.131	0.489 ± 0.623	0.469 ± 1.245	0.486 ± 1.102	0.497 ± 0.875	0.475 ± 0.788	0.469 ± 0.835	0.487 ± 0.872	0.472 ± 0.851
Dose uniformity (mg)		251 ± 1.285	249 ± 0.988	247 ± 1.018	164 ± 0.932	166 ± 0892	167 ± 0.854	123 ± 0.689	125 ± 1.323	126 ± 1.126
Mechanical strength (N)		99.75 ± 3.324	98.95 ± 2.755	92.55 ± 3.653	95.35 ± 3.352	93.55 ± 2.983	92.11 ± 2.652	88.95 ± 2.347	85.75 ± 2.583	82.45 ± 2.741
Friability (%)		1.011 ± 0.012	0.866 ± 0.025	1.775 ± 0.053	1.158 ± 0.025	1.313 ± 0.026	1.895 ± 0.035	1.230 ± 0.028	1.325 ± 0.035	1.954 ± 0.325

The standard deviation (SD) for n = 3.

**Table 4 ijms-25-05137-t004:** The main thermal characteristics of the components of the formulations determined from DTA and DTG curves.

Sample	Stage/DTA Characteristics	T_onset_, °C	T_peak_, °C	T_endset_, °C	ΔW, %	Residue, %
AV	I/endo	47	80	91	4.36	15.74
	II/endo exo	292	361	378	79.90	
CHT	I/endo	53	73	105	12.21	17.81
	II/exo	277	302	325	42.72	
	III/exo	467	509	700	26.56	
CLZ	I/endo	234	292	305	95.60	4.40
KOL	I/endo	45	55	77	3.75	9.49
	II/endo exo	320	343	361	54.27	
	III/exo	416	447	473	32.49	
ST	I/endo	50	63	113	2.59	7.05
	II/endo	330	386	410	90.36	

**Table 5 ijms-25-05137-t005:** The main thermal characteristics of the formulations.

Sample	Stage/DTA Characteristics	T_onset_, °C	T_peak_, °C	T_endset_, °C	ΔW, %	Residue, %
F1b	I/endo	45	58	85	1.50	9.90
	II/endo	242	279	300	53.08	
	III/exo	327	338	359	21.51	
	IV/exo	428	442	479	14.01	
F2b	I/endo	47	55	79	1.16	8.04
	II/endo	232	282	300	47.09	
	III/exo	326	336	357	25.52	
	IV/exo	430	440	481	18.19	
F3b	I/endo	45	51	74	0.80	11.04
	II/endo	235	283	303	47.52	
	III/exo	326	338	359	24.58	
	IV/exo	414	444	473	16.06	

**Table 6 ijms-25-05137-t006:** Values of f1 and f2 factors for the matrix tablet formulations.

Reference Formulations	Test Formulations	Difference Factor f1	Similarity Factor f2
F1a	F2a	41.8503	38.3546
	F3a	48.0360	30.0462
F1b	F2b	33.7543	57.9562
	F3b	35.8516	51.5645
F1c	F2c	52.1065	19.2135
	F3c	55.5045	19.8517

**Table 7 ijms-25-05137-t007:** Release parameters of chlorzoxazone release from the three formulations.

Sample Name	Q_max_ [%]	T_1/2_ [hours]
F1b	95.76	5.14
F2b	98.75	4.60
F3b	86.15	5.10

Q_max_ = maximum release amount; T_1/2_ = half release time.

**Table 8 ijms-25-05137-t008:** Kinetic parameters of chlorzoxazone release from investigated samples.

Sample Name	n	R^2^_n_	k [hour^–n^]	R^2^_k_
First step of kinetic release profile (0–2h)—pH 1.2				
F1b	0.114	0.966	0.190	0.999
F2b	0.077	0.604	0.197	0.998
F3b	0.196	0.965	0.168	0.998
Second step of kinetic release profile (2–36h)—pH 6.8				
F1b	0.708	0.982	0.151	0.990
F2b	0.673	0.977	0.170	0.983
F3b	0.577	0.983	0.167	0.991

n = release exponent, k = release rate constant, R^2^_n_ and R^2^_k_ = correlation coefficients corresponding to the slope obtained for the determination of n and k.

## Data Availability

Available on reasonable demand.

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
