# Peer review of "New Hydrophilic Matrix Tablets for the Controlled Released of Chlorzoxazone"

_ijms, 2024, doi:10.3390/ijms25105137_

Round 1

Reviewer 1 Report

Comments and Suggestions for Authors

The manuscript ijms-2976801 “Formulation and Evaluation of Hydrophilic Matrix Tablets for Controlled Released Chlorzoxazone Delivery” by Andreea Creteanu et al. describes the development of oral matrix tablets based on Kollidon®SR and chitosan to improve oral bioavailability of chlorzoxazone. 

The authors used modern research methods. The used references are reasonable.  

Questions and comments:

  1. Information about matrix tablets should be added to the Introduction. 

  2. Lines 55 - 77 - Contradictory data are given from which it is unclear whether the CLZ is well or poorly absorbed in the GI tract.

  3. Lines 78 - 79 - Information about BCS Class II drugs should be added, primarily their pharmaceutical solubility and intestinal permeability levels.

  4. Figure 3 - The formula of chitosan is failed, with extra CH2 groups present at the glycosidic bonds. 

  5. Lines 114 - 116 Information should be moved in Materials and Methods.

  6. Lines 121 - 129 - What are the advantages of matrix tablets for pediatric and geriatric practice versus traditional tablets.

  7. Lines 132 - 138 - It is unclear what the composition of the F formulations are. Add a reference to Section 3.2.

  8. Section 3.1 - Characteristics of chitosan (molecular weight and degree of deacetylation) should be added.

Author Response

Comment 1:Information about   matrix  tablets should be added to the Introduction.       

Response1: The auxiliary substances were formulated in constant concentrations: Aerosil® 1 (w/w)% , Magnesium Stearate 0.5 (w/w) % and Avicel® up to 100 (w/w) %.

Comment2: Lines 55 - 77 - Contradictory data are given from which it is unclear whether the CLZ is well or poorly absorbed in the GI tract.

Response2:CLZ has a Pka value of 3.3, a lower aqueous solubility of 0.2–0.3 mg/ml and high membrane permeability, therefore dissolution in gastrointestinal fluids is the limiting step for its oral bioavailability [7’

Ahmed Z. Abdel-Azeem, Atef A. Abdel-Hafez, Gamal   S.         El-Karamany,   Hassan H.        Farag. Chlorzoxazone esters of some non-steroidal anti- inflammatory (NSAI) carboxylic acids as mutual prodrugs: Design, synthesis, pharmacological investigations and docking studies. Bioorganic & Medicinal Chemistry. 2009, 17 (10), 3665-3670. Chlorzoxazone is rapidly absorbed from the gastrointestinal tract.                       Therapeutically active plasma concentrations are maintained for 3-4 hours. The solubility of a drug influence the choice of formulation for oral administration, dissolution, and absorption from the digestive tube and it is a suitable candidate for formulation as a gastroretentive dose form.

Comment3:.Lines 78 - 79 - Information about BCS Class II drugs should be added, primarily   their   pharmaceutical

solubility and intestinal permeability levels.      

Response3:It is considered a Class II drug (according to the Biopharmaceutical Classification System) because it has low solubility and high membrane permeability.

Comment 4: Figure 3 - The formula of chitosan is failed,

with     extra    CH2     groups  present at         the glycosidic bonds.   

Response4: Figure 3 redone.

Comment 5: Lines 114 - 116 Information should be

moved in Materials and Methods.         

Response5: They have been moved

 Comment 6:.Lines 121 - 129 - What are the advantages of matrix tablets for pediatric and geriatric practice versus traditional tablets.        

Response 6: The present matrix tablets for pedriatric and geriatric practice versus traditional tablets is aimed at to improve the patient compliance, the number of administered doses is reduced,

to increase the solubility by preparing sustained release tablets.

Comment 7: Lines 132 - 138 - It is unclear what the composition of the F formulations are. Add a

reference to Section 3.2.

Response 7: Moved table 8 from Section 3.2. It is now Table 1

Comment8: Section 3.1 - Characteristics of chitosan (molecular  weight  and      degree  of

deacetylation) should be added.

Response8: Chitosan characteristics have been mentioned

Reviewer 2 Report

Comments and Suggestions for Authors

The manuscript by Creteanu A. et al, entitled Formulation and Evaluation of Hydrophilic Matrix Tablets for Controlled Released Chlorzoxazone Delivery, presents an interesting and useful study on the development of some chlorzoxazone controlled release tablets.

The study is well conducted and structured, but the main problem of the manuscript is the English language (as already evident from the title) which needs extensive revision. Many sentences are not well formulated and are difficult to understand.

Some examples only in the introduction section:

Lines 59-60: The associated muscle is a decrease in skeletal muscular spasm accompanied by enhanced mobility and pain alleviation.

Lines 65-66: Most participants may attain peak levels of CLZ in 1 to 2 hours following oral administration, with blood levels typically noticed in persons during the first 30 minutes.

Lines 81-82: Any active substance is gastrointestinal absorption is severely limited by its low water solubility

Lines 112-115: CHT with a degree of deacetylation from 51% to 65% only increases the absorption  of active substances hydrophores with high molecular weight[16,17].

Avicel® PH (AV): Microcrystalline Cellulose.

Aerosil® (A): hydrophilic fumed silica with a specific surface area of 200 m² g-1 .

Magnesium Stearate (ST) has also been used.

Lines 135-136: The results obtained for the Hausner ratio and Carr index set the F1a, F1b, F2a, F2b, F3a, and F3b formulations into the group of powders with sound and excellent flow.

Concerning the methodology of the study, some questions need to be clarified:

-        Why did the authors chose the 50, 33 and 25 mg as the CLZ concentrations in the tablets? As already presented in the introduction, the minimum active concentration of CLZ is 250 mg.

-        It is not usual to vary the active ingredient concentration when changing the mass ratio between it and the excipients. The authors did not report to a therapeutic dosage to obtain the tablets, but only set the final mass of the tablets at 100 mg. Then, they considered varying the amounts of the excipients to the detriment of the API. This leads the study only to state of the art, not to a product with therapeutic benefit.

-        The Hausner ratio and the Carr index are indicators of flowability and compressibility, not compatibility

-        Section 3.3. must be rewritten. In the present form, the tablets were obtained first, then the powders for direct compression (probably the Erweka AR 403 device is a mixer), and in the end, again, is a compression method.

-        Regarding the pharmacotechnical properties of the tablets, I don’t think is suitable to use the Romanian Pharmacopoeia, Xth Edition, as a standard, considering that, in EU, all tablets must have characteristics in accordance with the European Pharmacopoeia specifications.

-        Equation 4 is not correct

-        Why didn’t the authors use the CLZ tablets on the market for comparison in the dissolution studies? The difference factor f1 and the similarity factor f2 would have been more relevant.

Regarding the Results section:

- Besides the inappropriate English language, some discussions must be added. The authors must not only present the obtained results, but also evaluate them in comparison to the similar findings mentioned in other studies and the data existed on the excipients properties that influence the behavior of the tablets.

-        Tables 1 and 2 are inappropriate and must be arranged clearly

-        Figure 7 requires more explanations

-        What are 1, 2 and 3 representing in Figure 8b?

-        The numbers of the figures mentioned in lines 354-364 are not correct

-        Please refer to the same abbreviations of F1c, F2c, and F3c in all text

-        Please insert the SD lines in Figure 13

Comments on the Quality of English Language

Moderate editing of English language required

Author Response

Comments and Suggestions for Authors

The manuscript by Creteanu A. et al, entitled Formulation and Evaluation of Hydrophilic Matrix Tablets for Controlled Released Chlorzoxazone Delivery, presents an interesting and useful study on the development of some chlorzoxazone controlled release tablets.

The study is well conducted and structured, but the main problem of the manuscript is the English language Many sentences are not well formulated and are difficult to understand.

Comment 1: (as already evident from the title) which needs extensive revision.     

Response1:New Hydrophilic Matrix Tablets for the Controlled Released of the Chlorzoxazone

Some examples only in the introduction section:

Comment2: Lines 59-60: The associated muscle is a decrease in skeletal muscular spasm accompanied by enhanced mobility and pain alleviation.        

Response2: CLZ  is a centrally acting musculoskeletal relaxant with sedative properties. CLZ constrains the multisynaptic reflex arcs producing from the spinal cord and subcortical regions of the brain which prolong the cause for maintaining the skeletal muscle spasm . CLZ is also used for acute pain relief and headache due to muscle contraction.[ref]

Comment3:. Lines 65-66:Most participants may attain peak levels of CLZ in 1 to 2 hours following oral administration, with blood levels typically noticed in persons during the first 30 minutes. 

Response3: Therapeutically active plasma concentrations are maintained for 3-4 hours. The mean plasma half-life is 1 hour. CLZ is rapidly metabolized in the liver by the CYP2E isozyme of the citrochrome P450.CLZ is eliminated renally, mainly in conjugated form (as glucuronide) and <1% in unchanged form [ref].

Commetn4: Lines 81-82:Any active substance is gastrointestinal absorption is severely limited by its low water solubility        

Response4:The release of the active substance from the matrix modified-release tablets is dependent on its degree of solubility in the dissolution medium as well as on the composition of the matrix forming.

Comment5: Lines 112-115: CHT with a degree of deacetylation from 51% to 65% only increases the absorption  of active substances hydrophores with high molecular weight[16,17].

Reponse5: corrected

Comment6:.Avicel® PH (AV): Microcrystalline Cellulose. 

Response6: corrected

Comment 7:.Aerosil® (A): hydrophilic fumed silica with a specific surface area of 200 m² g-1 Response7:corrected

Comment8:.Magnesium Stearate (ST) has also been used.  

Response8: corrected

Comment 9: Lines 135-136: The results obtained for the Hausner ratio and Carr index set the F1a, F1b, F2a, F2b, F3a, and F3b formulations into the group of powders with sound and excellent flow.

Response9: corrected

Concerning the methodology of the study, some questions need to be clarified:

Comment10: Why did the authors chose the 50, 33 and 25 mg as the CLZ concentrations in the tablets? As already presented in the introduction, the minimum active concentration of CLZ is 250 mg.         

Response10: In 100 (w/w)%  powder mixture 50 (w/w) %  , 33 (w/w )%  or 25( w/w )%  represent CLZ.

The proportion of KOL (20-40 w/w %  ) and CHT 3-5 (w/w %  ) was varied, the ST and AE coanstant values  and the AV completes at 100 w/w %  .

These proportions were taken into account when preparing the tablets, but the average weiht per tablet is 500 mg.A matrix tablet can contain 250 mg CLZ (F1), 165 mg CLZ ()F2) or 125mg CLZ (F3).

Comment 11: It is not usual to vary the active ingredient concentration when changing the mass ratio between it and the excipients. The authors did not report to a therapeutic dosage to obtain the tablets, but only set the final mass of the tablets at 100 mg. Then, they considered varying the amounts of the excipients to the detriment of the API. This leads the study only to state of the art, not to a product with therapeutic benefit. 

Response11: The influence of KOL and CHT on the quality of the matrix is followed.  The results showed that the formulations containing KOL in 20-30 (w/w)% concentration presented a good flow, whereas the increase in KOL concentration above 40 (w/w) % influenced negatively the flow properties.

It has been also found that increasing the CHT concentration in the formulation resulted in a decrease in the rate of CLZ release from the matrix tablets.

Comment12: The Hausner ratio and the Carr index are indicators of flowability and compressibility, not compatibility  

Response12: corrected

Comment 13: Section 3.3. must be rewritten. In the present form, the tablets were obtained first, then the powders for direct compression (probably the Erweka AR 403 device is a mixer), and in the end, again, is a compression method. 

Response13:The mixture of powders corresponding to the formulation of matrix tablets with CLZ were mixed in an Erweka AR 403 mixer (Erweka GmbH, Heusenstamm, Germany) with a rotation speed of 400 rpm for 5 minutes, after which they were sieved using an EM-8

electromagnetic sieve (Erweka GmbH, Heusenstamm, Germany), then they were subjected to compression directly to the Korsch EK0 compression machine (Korsch AG, Berlin, Germany).

Comment14: Regarding the pharmacotechnical properties of the tablets, I don’t think is suitable to use the Romanian Pharmacopoeia, Xth Edition, as a standard, considering that, in EU, all tablets must have characteristics in accordance with the European Pharmacopoeia specifications.     

Response14: The change was made: European Pharmacopoeia, 8th ed. 2014, 288.

Comment15: Equation 4 is not correct          

Response15: corrected

Comment16. Why didn’t the authors use the CLZ tablets on the market for comparison in the dissolution studies? The difference factor f1 and the similarity factor f2 would have been more relevant.       

Response 16: corrected - Formulations F2c and F3c with a maximum concentration of KOL 40 (w/w )% and CHT 7(w/w) % have inappropriate values of f1 and f2, demonstrating the influence of KOL and CHT. As the quantity of KOL 20-30 (w/w) % and CHT 3-5(w/w) % decreases in the matrix, f1 and f2 have corresponding values and the quantity of CLZ yielded increases

Regarding the Results section:

Comment17: Besides the inappropriate English language, some discussions must be added. The authors must not only present the obtained results, but also evaluate them in comparison to the similar findings mentioned in other studies and the data existed on the excipients properties that influence the behavior of the tablets.  

Response17:In sections related to FTIR, XRD, TG, DTA,DSC, the comparison with other studies and data is made

Comment18: Tables 1 and 2 are inappropriate and must be arranged clearly           

Response18: corrected

Comment19. Figure 7 requires more explanations    

Response 19:corrected

Comment 20. What are 1, 2 and 3 representing in Figure 8b?          

Response20: corrected

Comment 21. The numbers of the figures mentioned in lines 354-364 are not correct

Response 21:corrected

Comment 22. Please refer to the same abbreviations of F1c, F2c, and F3c in all text

Response22:corrected

Comment 23.Please insert the SD lines in Figure 13

Response23:corrected

Comments on the Quality of English Language

Response: Moderate editing of English language required done

Round 2

Reviewer 1 Report

Comments and Suggestions for Authors

The manuscript may be accepted for publication, however, some comments remain.

1) Line 141 - Mw = 10,000-1,000,000 Daltons replace with Mw = 10,000 - 1,000,000

2) Figure 3. - The terminal units of the chitosan macrochain must be removed.

3) Lines 161 - 163 - molecular weight of chitosan should be added.

Author Response

Comment 1: Line 141 - Mw = 10,000-1,000,000 Daltons replace with Mw = 10,000 - 1,000,000

Response1: corrected as suggested

Comment 2: Figure 3. - The terminal units of the chitosan macrochain must be removed.

Response 2: Figure 3 was corrected as suggested

Comment 3: Lines 161 - 163 - molecular weight of chitosan should be added.

Response 3: Molecular weight was added as suggested.

Reviewer 2 Report

Comments and Suggestions for Authors

Dear authors,

Thank you for all the answers. Still, English language requiers editing. The correct title would be: New Hydrophilic Matrix Tablets for the Controlled Release of Chlorzoxazone. Still, the tables need to be adjusted.

Comments on the Quality of English Language

English language requiers editing.

Author Response

Comment1: The correct title would be: New Hydrophilic Matrix Tablets for the Controlled Release of Chlorzoxazone.

Response1: The title was corrected as suggested

Comment 2: Still, the tables need to be adjusted.

Response2: The tables ( 1 and 2) have been adjusted

The English language  was corrected

Thank you for your review and your valuable comments